# CMA-ES with Optimal Covariance Update and Storage Complexity

**Oswin Krause**
Dept. of Computer Science
University of Copenhagen
Copenhagen, Denmark
oswin.krause@di.ku.dk

**Dídac R. Arbonès**
Dept. of Computer Science
University of Copenhagen
Copenhagen, Denmark
didac@di.ku.dk

**Christian Igel**
Dept. of Computer Science
University of Copenhagen
Copenhagen, Denmark
igel@di.ku.dk

## Abstract

The covariance matrix adaptation evolution strategy (CMA-ES) is arguably one of the most powerful real-valued derivative-free optimization algorithms, finding many applications in machine learning. The CMA-ES is a Monte Carlo method, sampling from a sequence of multi-variate Gaussian distributions. Given the function values at the sampled points, updating and storing the covariance matrix dominates the time and space complexity in each iteration of the algorithm. We propose a numerically stable quadratic-time covariance matrix update scheme with minimal memory requirements based on maintaining triangular Cholesky factors. This requires a modification of the cumulative step-size adaption (CSA) mechanism in the CMA-ES, in which we replace the inverse of the square root of the covariance matrix by the inverse of the triangular Cholesky factor. Because the triangular Cholesky factor changes smoothly with the matrix square root, this modification does not change the behavior of the CMA-ES in terms of required objective function evaluations as verified empirically. Thus, the described algorithm can and should replace the standard CMA-ES if updating and storing the covariance matrix matters.

## 1   Introduction

The *covariance matrix adaptation evolution strategy*, CMA-ES [Hansen and Ostermeier, 2001], is recognized as one of the most competitive derivative-free algorithms for real-valued optimization [Beyer, 2007; Eiben and Smith, 2015]. The algorithm has been successfully applied in many unbiased performance comparisons and numerous real-world applications. In machine learning, it is mainly used for direct policy search in reinforcement learning and hyperparameter tuning in supervised learning (e.g., see Gomez *et al.* [2008]; Heidrich-Meisner and Igel [2009a,b]; Igel [2010], and references therein).

The CMA-ES is a Monte Carlo method for optimizing functions $f : \mathbb{R}^d \to \mathbb{R}$. The objective function $f$ does not need to be continuous and can be multi-modal, constrained, and disturbed by noise. In each iteration, the CMA-ES samples from a $d$-dimensional multivariate normal distribution, the *search distribution*, and ranks the sampled points according to their objective function values. The mean and the covariance matrix of the search distribution are then adapted based on the ranked points. Given the ranking of the sampled points, the runtime of one CMA-ES iteration is $\omega(d^2)$ because the square root of the covariance matrix is required, which is typically computed by an eigenvalue decomposition. If the objective function can be evaluated efficiently and/or $d$ is large, the computation of the matrix square root can easily dominate the runtime of the optimization process.

Various strategies have been proposed to address this problem. The basic approach for reducing the runtime is to perform an update of the matrix only every $\tau \in \Omega(d)$ steps [Hansen and Ostermeier,

1996, 2001], effectively reducing the time complexity to $O(d^2)$. However, this forces the algorithm to use outdated matrices during most iterations and can increase the amount of function evaluations. Furthermore, it leads to an uneven distribution of computation time over the iterations. Another approach is to restrict the model complexity of the search distribution [Poland and Zell, 2001; Ros and Hansen, 2008; Sun *et al.*, 2013; Akimoto *et al.*, 2014; Loshchilov, 2014, 2015], for example, to consider only diagonal matrices [Ros and Hansen, 2008]. However, this can lead to a drastic increase in function evaluations needed to approximate the optimum if the objective function is not compatible with the restriction, for example, when optimizing highly non-separable problems while only adapting the diagonal of the covariance matrix [Omidvar and Li, 2011]. More recently, methods were proposed that update the Cholesky factor of the covariance matrix instead of the covariance matrix itself [Suttorp *et al.*, 2009; Krause and Igel, 2015]. This works well for some CMA-ES variations (e.g., the (1+1)-CMA-ES and the multi-objective MO-CMA-ES [Suttorp *et al.*, 2009; Krause and Igel, 2015; Bringmann *et al.*, 2013]), however, the original CMA-ES relies on the matrix square root, which cannot be replaced one-to-one by a Cholesky factor.

In the following, we explore the use of the triangular Cholesky factorization instead of the square root in the standard CMA-ES. In contrast to previous attempts in this direction, we present an approach that comes with a theoretical justification for why it does not deteriorate the algorithm's performance. This approach leads to the optimal asymptotic storage and runtime complexity when adaptation of the full covariance matrix is required, as is the case for non-separable ill-conditioned problems. Our CMA-ES variant, referred to as Cholesky-CMA-ES, reduces the runtime complexity of the algorithm with no significant change in the number of objective function evaluations. It also reduces the memory footprint of the algorithm.

Section 2 briefly describes the original CMA-ES algorithm (for details we refer Hansen [2015]). In section 3 we propose our new method for approximating the step-size adaptation. We give a theoretical justification for the convergence of the new algorithm. We provide empirical performance results comparing the original CMA-ES with the new Cholesky-CMA-ES using various benchmark functions in section 4. Finally, we discuss our results and draw our conclusions.

## 2 Background

Before we briefly describe the CMA-ES to fix our notation, we discuss some basic properties of using a Cholesky decomposition to sample from a multi-variate Gaussian distribution. Sampling from a $d$-dimensional multi-variate normal distribution $\mathcal{N}(\boldsymbol{m}, \Sigma)$, $\boldsymbol{m} \in \mathbb{R}^d$, $\Sigma \in \mathbb{R}^{d \times d}$ is usually done using a decomposition of the covariance matrix $\Sigma$. This could be the square root of the matrix $\Sigma = HH \in \mathbb{R}^{d \times d}$ or a lower triangular Cholesky factorization $\Sigma = AA^T$, which is related to the square root by the QR-decomposition $H = AE$ where $E$ is an orthogonal matrix. We can sample a point $\boldsymbol{x}$ from $\mathcal{N}(\boldsymbol{m}, \Sigma)$ using a sample $\boldsymbol{z} \sim \mathcal{N}(\boldsymbol{0}, \mathrm{I})$ by $\boldsymbol{x} = H\boldsymbol{z} + \boldsymbol{m} = AE\boldsymbol{z} + \boldsymbol{m} = A\boldsymbol{y} + \boldsymbol{m}$, where we set $\boldsymbol{y} = E\boldsymbol{z}$. We have $\boldsymbol{y} \sim \mathcal{N}(\boldsymbol{0}, \mathrm{I})$ since $E$ is orthogonal. Thus, as long as we are only interested in the value of $\boldsymbol{x}$ and do not need $\boldsymbol{y}$, we can sample using the Cholesky factor instead of the matrix square root.

### 2.1 CMA-ES

The CMA-ES has been proposed by Hansen and Ostermeier [1996, 2001] and its most recent version is described by Hansen [2015]. In the $t$th iteration of the algorithm, the CMA-ES samples $\lambda$ points from a multivariate normal distribution $\mathcal{N}(\boldsymbol{m}_t, \sigma_t^2 \cdot C_t)$, evaluates the objective function $f$ at these points, and adapts the parameters $C_t \in \mathbb{R}^{d \times d}$, $\boldsymbol{m}_t \in \mathbb{R}^d$, and $\sigma_t \in \mathbb{R}^+$. In the following, we present the update procedure in a slightly simplified form (for didactic reasons, we refer to Hansen [2015] for the details). All parameters ($\mu$, $\lambda$, $\boldsymbol{\omega}$, $c_\sigma$, $d_\sigma$, $c_c$, $c_1$, $c_\mu$) are set to their default values [Hansen, 2015, Table 1].

For a minimization task, the $\lambda$ points are ranked by function value such that $f(\boldsymbol{x}_{1,t}) \leq f(\boldsymbol{x}_{2,t}) \leq \cdots \leq f(\boldsymbol{x}_{\lambda,t})$. The distribution mean is set to the weighted average $\boldsymbol{m}_{t+1} = \sum_{i=1}^{\mu} \omega_i \boldsymbol{x}_{i,t}$. The weights depend only on the ranking, not on the function values directly. This renders the algorithm invariant under order-preserving transformation of the objective function. Points with smaller ranks (i.e., better objective function values) are given a larger weight $\omega_i$ with $\sum_{i=1}^{\lambda} \omega_i = 1$. The weights are zero for ranks larger than $\mu < \lambda$, which is typically $\mu = \lambda/2$. Thus, points with function values worse than the median do not enter the adaptation process of the parameters. The covariance matrix

is updated using two terms, a rank-1 and a rank-$\mu$ update. For the rank-1 update, a long term average of the changes of $\boldsymbol{m}_t$ is maintained

$$\boldsymbol{p}_{c,t+1} = (1 - c_c)\boldsymbol{p}_{c,t} + \sqrt{c_c(2 - c_c)\mu_{\text{eff}}}\frac{\boldsymbol{m}_{t+1} - \boldsymbol{m}_t}{\sigma_t} \quad , \tag{1}$$

where $\mu_{\text{eff}} = 1/\sum_{i=1}^{\mu} \omega_i^2$ is the effective sample size given the weights. Note that $\boldsymbol{p}_{c,t}$ is large when the algorithm performs steps in the same direction, while it becomes small when the algorithm performs steps in alternating directions.[1] The rank-$\mu$ update estimates the covariance of the weighted steps $\boldsymbol{x}_{i,t} - \boldsymbol{m}_t$, $1 \le i \le \mu$. Combining rank-1 and rank-$\mu$ update gives the final update rule for $C_t$, which can be motivated by principles from information geometry [Akimoto *et al.*, 2012]:

$$C_{t+1} = (1 - c_1 - c_\mu)C_t + c_1\boldsymbol{p}_{c,t+1}\boldsymbol{p}_{c,t+1}^T + \frac{c_\mu}{\sigma_t^2} \sum_{i=1}^{\mu} \omega_i \left(\boldsymbol{x}_{i,t} - \boldsymbol{m}_t\right)\left(\boldsymbol{x}_{i,t} - \boldsymbol{m}_t\right)^T \tag{2}$$

So far, the update is (apart from initialization) invariant under affine linear transformations (i.e., $\boldsymbol{x} \mapsto B\boldsymbol{x} + \boldsymbol{b}$, $B \in \text{GL}(d, \mathbb{R})$).

The update of the global step-size parameter $\sigma_t$ is based on the *cumulative step-size adaptation* algorithm (CSA). It measures the correlation of successive steps in a normalized coordinate system. The goal is to adapt $\sigma_t$ such that the steps of the algorithm become uncorrelated. Under the assumption that uncorrelated steps are standard normally distributed, a carefully designed long term average over the steps should have the same expected length as a $\chi$-distributed random variable, denoted by $\mathbb{E}\{\chi\}$. The long term average has the form

$$\boldsymbol{p}_{\sigma,t+1} = (1 - c_\sigma)\boldsymbol{p}_{\sigma,t} + \sqrt{c_\sigma(2 - c_\sigma)\mu_{\text{eff}}}\, C_t^{-1/2}\frac{\boldsymbol{m}_{t+1} - \boldsymbol{m}_t}{\sigma_t} \tag{3}$$

with $\boldsymbol{p}_{\sigma,1} = 0$. The normalization by the factor $C_t^{-1/2}$ is the main difference between equations (1) and (3). It is important because it corrects for a change of $C_t$ between iterations. Without this correction, it is difficult to measure correlations accurately in the un-normalized coordinate system. For the update, the length of $\boldsymbol{p}_{\sigma,t+1}$ is compared to the expected length $\mathbb{E}\{\chi\}$ and $\sigma_t$ is changed depending on whether the average step taken is longer or shorter than expected:

$$\sigma_{t+1} = \sigma_t \exp\left(\frac{c_\sigma}{d_\sigma}\left(\frac{\|\boldsymbol{p}_{\sigma,t+1}\|}{\mathbb{E}\{\chi\}} - 1\right)\right) \tag{4}$$

This update is *not* proven to preserve invariance under affine linear transformations [Auger, 2015], and it is it conjectured that it does not.

## 3  Cholesky-CMA-ES

In general, computing the matrix square root or the Cholesky factor from an $n \times n$ matrix has time complexity $\omega(d^2)$ (i.e., scales worse than quadratically). To reduce this complexity, Suttorp *et al.* [2009] have suggested to replace the process of updating the covariance matrix and decomposing it afterwards by updates directly operating on the decomposition (i.e., the covariance matrix is never computed and stored explicitly, only its factorization is maintained). Krause and Igel [2015] have shown that the update of $C_t$ in equation (2) can be rewritten as a quadratic-time update of its triangular Cholesky factor $A_t$ with $C_t = A_t A_t^T$. They consider the special case $\mu = \lambda = 1$. We propose to extend this update to the standard CMA-ES, which leads to a runtime $O(\mu d^2)$. As typically $\mu = O(\log(d))$, this gives a large speed-up compared to the explicit recomputation of the Cholesky factor or the inverse of the covariance matrix.

Unfortunately, the fast Cholesky update can not be applied directly to the original CMA-ES. To see this, consider the term $\boldsymbol{s}_t = C_t^{-1/2}(\boldsymbol{m}_{t+1} - \boldsymbol{m}_t)$ in equation (3). Rewriting $\boldsymbol{p}_{\sigma,t+1}$ in terms of $\boldsymbol{s}_t$ in a non-recursive fashion, we obtain

$$\boldsymbol{p}_{\sigma,t+1} = \sqrt{c_\sigma(2 - c_\sigma)\mu_{\text{eff}}} \sum_{k=1}^{t} \frac{(1 - c_\sigma)^{t-k}}{\sigma_k} \boldsymbol{s}_k \quad .$$

**Algorithm 1:** The Cholesky-CMA-ES.

---

**input :** $\lambda$, $\mu$, $\boldsymbol{m}_1$, $\omega_{i=1\ldots\mu}$, $c_\sigma$, $d_\sigma$, $c_c$, $c_1$ and $c_\mu$
$A_1 = \mathrm{I}, p_{c,1} = \boldsymbol{0}, p_{\sigma,1} = \boldsymbol{0}$
**for** $t = 1, 2, \ldots$ **do**
$\quad$ **for** $i = 1, \ldots, \lambda$ **do**
$\quad\quad \lfloor\; \boldsymbol{x}_{i,t} = \sigma_t A_t \boldsymbol{y}_{i,t} + \boldsymbol{m}_t,\; \boldsymbol{y}_{i,t} \sim \mathcal{N}(\boldsymbol{0}, \mathrm{I})$
$\quad$ Sort $\boldsymbol{x}_{i,t}, i = 1, \ldots, \lambda$ increasing by $f(\boldsymbol{x}_{i,t})$
$\quad \boldsymbol{m}_{t+1} = \sum_{i=1}^{\mu} \omega_i \boldsymbol{x}_{i,t}$
$\quad \boldsymbol{p}_{c,t+1} = (1 - c_c)\boldsymbol{p}_{c,t} + \sqrt{c_c(2 - c_c)\mu_{\mathrm{eff}}}\frac{\boldsymbol{m}_{t+1} - \boldsymbol{m}_t}{\sigma_t}$
$\quad$ // Apply formula (2) to $A_t$
$\quad A_{t+1} \leftarrow \sqrt{1 - c_1 - c_\mu} A_t$
$\quad A_{t+1} \leftarrow \mathrm{rankOneUpdate}(A_{t+1}, c_1, p_{c,t+1})$
$\quad$ **for** $i = 1, \ldots, \mu$ **do**
$\quad\quad \lfloor\; A_{t+1} \leftarrow \mathrm{rankOneUpdate}(A_{t+1}, c_\mu\omega_i, \frac{\boldsymbol{x}_{i,t} - \boldsymbol{m}_t}{\sigma_t})$
$\quad$ // Update $\sigma$ using $\hat{\boldsymbol{s}}_k$ as in (5)
$\quad \boldsymbol{p}_{\sigma,t+1} = (1 - c_\sigma)\boldsymbol{p}_{\sigma,t} + \sqrt{c_\sigma(2 - c_\sigma)\mu_{\mathrm{eff}}}A_t^{-1}\frac{\boldsymbol{m}_{t+1} - \boldsymbol{m}_t}{\sigma_t}$
$\quad \sigma_{t+1} = \sigma_t \exp\left(\frac{c_\sigma}{d_\sigma}\left(\frac{\|\boldsymbol{p}_{\sigma,t+1}\|}{\mathbb{E}\{\chi\}} - 1\right)\right)$

---

**Algorithm 2:** rankOneUpdate$(A, \beta, \boldsymbol{v})$

---

**input :** Cholesky factor $A \in \mathbb{R}^{d \times d}$ of $C$, $\beta \in \mathbb{R}$, $\boldsymbol{v} \in \mathbb{R}^d$
**output :** Cholesky factor $A'$ of $C + \beta \boldsymbol{v}\boldsymbol{v}^T$
$\boldsymbol{\alpha} \leftarrow \boldsymbol{v}$
$b \leftarrow 1$
**for** $j = 1, \ldots, d$ **do**
$\quad A'_{jj} \leftarrow \sqrt{A_{jj}^2 + \frac{\beta}{b}\alpha_j^2}$
$\quad \gamma \leftarrow A_{jj}^2 b + \beta\alpha_j^2$
$\quad$ **for** $k = j + 1, \ldots, d$ **do**
$\quad\quad \alpha_k \leftarrow \alpha_k - \frac{\alpha_j}{A_{jj}}A_{kj}$
$\quad\quad A'_{kj} = \frac{A'_{jj}}{A_{jj}}A_{kj} + \frac{A'_{jj}\beta\alpha_j}{\gamma}\alpha_k$
$\quad b \leftarrow b + \beta\frac{\alpha_j^2}{A_{jj}^2}$

---

By the RQ-decomposition, we can find $C_t^{1/2} = A_t E_t$ with $E_t$ being an orthogonal matrix and $A_t$ lower triangular. When replacing $\boldsymbol{s}_t$ by $\hat{\boldsymbol{s}}_t = A_t^{-1}(\boldsymbol{m}_{t+1} - \boldsymbol{m}_t)$, we obtain

$$\boldsymbol{p}_{\sigma,t+1} = \sqrt{c_\sigma(2 - c_\sigma)\mu_{\mathrm{eff}}} \sum_{k=1}^{t} \frac{(1 - c_\sigma)^{t-k}}{\sigma_k} E_k^T \hat{\boldsymbol{s}}_k \; .$$

Thus, replacing $C_t^{-1/2}$ by $A_t^{-1}$ introduces a new random rotation matrix $E_t^T$, which changes in every iteration. Obtaining $E_t$ from $A_t$ can be achieved by the polar-decomposition, which is a cubic-time operation: currently there are no algorithms known that can update an existing polar decomposition from an updated Cholesky factor in less than cubic time. Thus, if our goal is to apply the fast Cholesky update, we have to perform the update without this correction factor

$$\boldsymbol{p}_{\sigma,t+1} \approx \sqrt{c_\sigma(2 - c_\sigma)\mu_{\mathrm{eff}}} \sum_{k=1}^{t} \frac{(1 - c_\sigma)^{t-k}}{\sigma_k} \hat{\boldsymbol{s}}_k \; . \tag{5}$$

This introduces some error, but we will show in the following that we can expect this error to be small and to decrease over time as the algorithm converges to the optimum. For this, we need the following result:

**Lemma 1.** *Consider the sequence of symmetric positive definite matrices $\bar{C}_{t=0}^{\infty}$ with $\bar{C}_t = C_t(\det C_t)^{-1/d}$. Assume that $\bar{C}_t \overset{t\to\infty}{\Longrightarrow} \bar{C}$ and that $\bar{C}$ is symmetric positive definite with $\det \bar{C} = 1$. Let $\bar{C}_t^{1/2} = \bar{A}_t E_t$ denote the RQ-decomposition of $\bar{C}_t^{1/2}$, where $E_t$ is orthogonal and $\bar{A}_t$ lower triangular. Then it holds $E_{t-1}^T E_t \overset{t\to\infty}{\Longrightarrow} I$.*

*Proof.* Let $\bar{C} = \bar{A}E$, the RQ-decomposition of $\bar{C}$. As $\det \bar{C} \neq 0$, this decomposition is unique. Because the RQ-decomposition is continuous, it maps convergent sequences to convergent sequences. Therefore $E_t \overset{t\to\infty}{\Longrightarrow} E$ and thus, $E_{t-1}^T E_t \overset{t\to\infty}{\Longrightarrow} E^T E = I$. $\qquad\square$

This result establishes that, when $C_t$ converges to a certain shape (but not necessary to a certain scaling), $A_t$ and thus $E_t$ will also converge (up to scaling). Thus, as we only need the norm of $\boldsymbol{p}_{\sigma,t+1}$, we can rotate the coordinate system and by multiplying with $E_t$ we obtain

$$\|\boldsymbol{p}_{\sigma,t+1}\| = \|E_t \boldsymbol{p}_{\sigma,t+1}\| = \sqrt{c_\sigma(2-c_\sigma)\mu_{\text{eff}}} \left\| \sum_{k=1}^{t} \frac{(1-c_\sigma)^{t-k}}{\sigma_k} E_t E_k^T \hat{\boldsymbol{s}}_k \right\| . \tag{6}$$

Therefore, if $E_t E_{t-1}^T \overset{t\to\infty}{\Longrightarrow} I$, the error in the norm will also vanish due to the exponential weighting in the summation. Note that this does not hold for any decomposition $C_t = B_t B_t^T$. If we do not constrain $B_t$ to be triangular and allow any matrix, we do not have a bijective mapping between $C_t$ and $B_t$ anymore and the introduction of $\frac{d(d-1)}{2}$ degrees of freedom (as, e.g., in the update proposed by Suttorp *et al.* [2009]) allows the creation of non-converging sequences of $E_t$ even for $C_t = \text{const.}$

As the CMA-ES is a randomized algorithm, we cannot assume convergence of $C_t$. However, in simplified algorithms the expectation of $C_t$ converges [Beyer, 2014]. Still, the reasoning behind Lemma 1 establishes that the error caused by replacing $\boldsymbol{s}_t$ by $\hat{\boldsymbol{s}}_t$ is small if $C_t$ changes slowly. Equation (6) establishes that the error depends only on the rotation of coordinate systems. As the mapping from $C_t$ to the triangular factor $A_t$ is one-to-one and smooth, the coordinate system changes in every step will be small – and because of the exponentially decaying weighting, only the last few coordinate systems matter at a particular time step $t$.

The Cholesky-CMA-ES algorithm is given in Algorithm 1. One can derive the algorithm from the standard CMA-ES by decomposing (2) into a number of rank-1 updates $C_{t+1} = (((\alpha C_t + \beta_1 \boldsymbol{v}_1 \boldsymbol{v}_1^T) + \beta_2 \boldsymbol{v}_2 \boldsymbol{v}_2^T)\dots)$ and applying them to the Cholesky factor using Algorithm 2.

**Properties of the update rule.** The $O(\mu d^2)$ complexity of the update in the Cholesky-CMA-ES is asymptotically optimal.[2] Apart from the theoretical guarantees, there are several additional advantages compared to approaches using a non-triangular Cholesky factorization (e.g., Suttorp *et al.* [2009]). First, as only triangular matrices have to be stored, the storage complexity is optimal. Second, the diagonal elements of a triangular Cholesky factor are the square roots of the eigenvalues of the factorized matrix, that is, we get the eigenvalues of the covariance matrix for free. These are important, for example, for monitoring the conditioning of the optimization problem and, in particular, to enforce lower bounds on the variances of $\sigma_t C_t$ projected on its principal components. Third, a triangular matrix can be inverted in quadratic time. Thus, we can efficiently compute $A_t^{-1}$ from $A_t$ when needed, instead of having two separate quadratic-time updates for $A_t^{-1}$ and $A_t$, which requires more memory and is prone to numerical instabilities.

## 4 Experiments and Results

**Experiments.** We compared the Cholesky-CMA-ES with other CMA-ES variants.[3] The reference CMA-ES implementation uses a delay strategy in which the matrix square root is computed every $\max\left\{1, \frac{1}{10d(c_1+c_\mu)}\right\}$ iterations [Hansen, 2015], which equals one for the dimensions considered

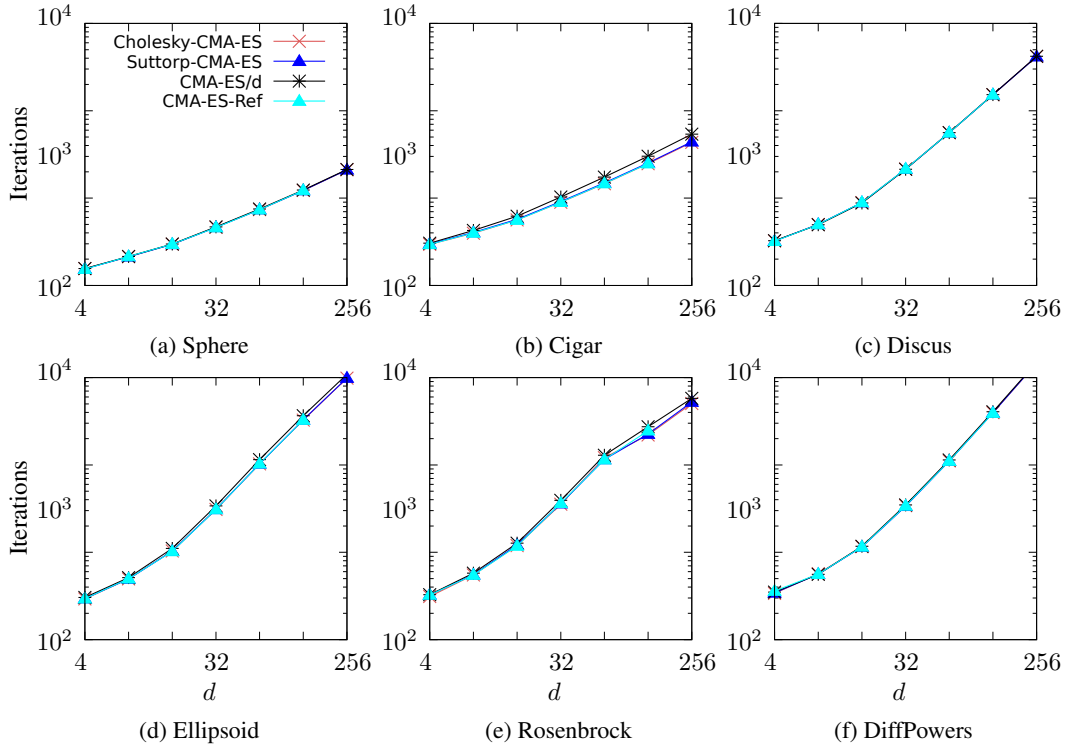

Figure 1: Function evaluations required to reach $f(\boldsymbol{x}) < 10^{-14}$ over problem dimensionality (medians of 100 trials). The graphs for CMA-ES-Ref and Cholesky-CMA-ES overlap.

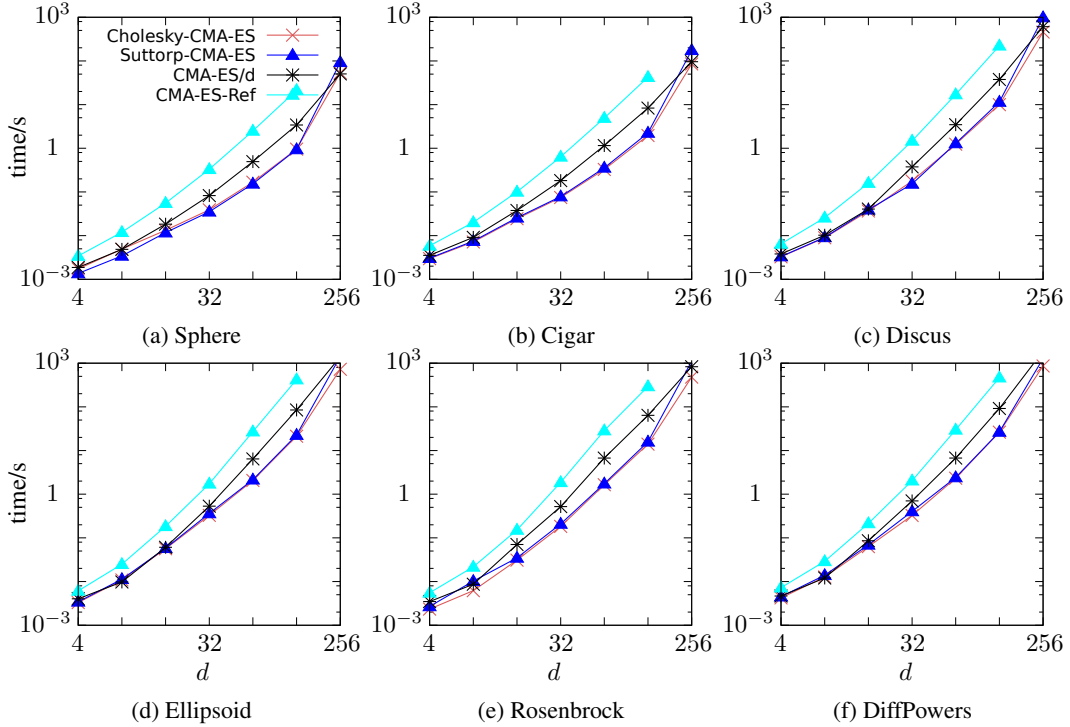

Figure 2: Runtime in seconds over problem dimensionality. Shown are medians of 100 trials. Note the logarithmic scaling on both axes.

| Name | $f(\boldsymbol{x})$ |
|---|---|
| Sphere | $\|\boldsymbol{x}\|^2$ |
| Rosenbrock | $\sum_{i=0}^{d-1} \left(100(x_{i+1} - x_i^2)^2 + (1-x_i)^2\right)$ |
| Discus | $x_0^2 + \sum_{i=1}^{d} 10^{-6} x_i^2$ |
| Cigar | $10^{-6}x_0^2 + \sum_{i=1}^{d} x_i^2$ |
| Ellipsoid | $\sum_{i=0}^{d} 10^{\frac{-6i}{d-1}} x_i^2$ |
| Different Powers | $\sum_{i=0}^{d} |x_i|^{\frac{2+10i}{d-1}}$ |

Table 1: Benchmark functions used in the experiments (additionally, a rotation matrix $B$ transforms the variables, $\boldsymbol{x} \mapsto B\boldsymbol{x}$)

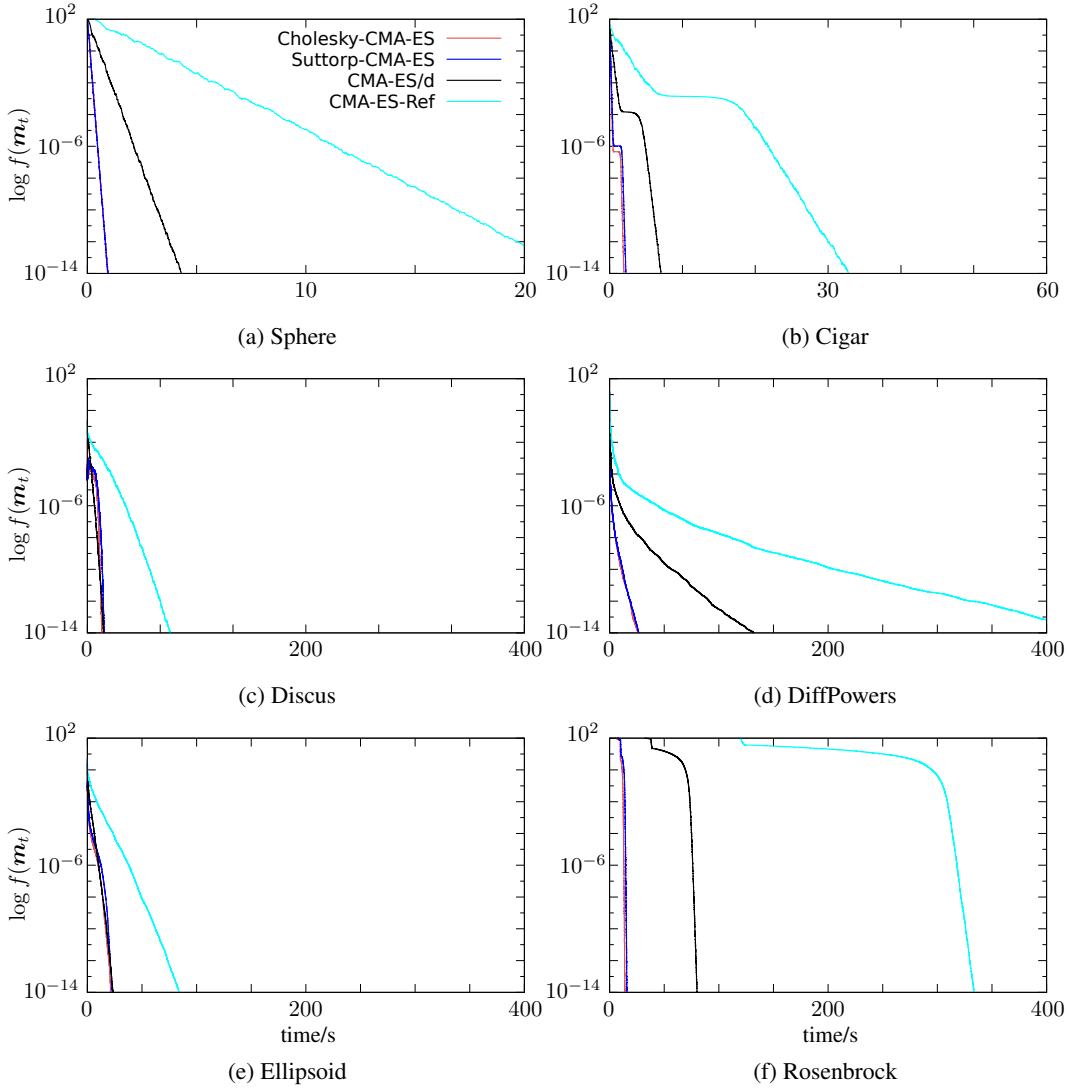

Figure 3: Function value evolution over time on the benchmark functions with $d = 128$. Shown are single runs, namely those with runtimes closest to the corresponding median runtimes.

in our experiments. We call this variant CMA-ES-Ref. As an alternative, we experimented with delaying the update for $d$ steps. We refer to this variant as CMA-ES/d. We also adapted the non-triangular Cholesky factor approach by Suttorp *et al.* [2009] to the state-of-the art implementation of the CMA-ES. We refer to the resulting algorithm as Suttorp-CMA-ES.

We considered standard benchmark functions for derivative-free optimization given in Table 1. Sphere is considered to show that on a spherical function the step size adaption does not behave differently; Cigar/Discus/Ellipsoid model functions with different convex shapes near the optimum; Rosenbrock tests learning a function with $d - 1$ bends, which lead to slowly converging covariance matrices in the optimization process; Diffpowers is an example of a function with arbitrarily bad conditioning.

To test rotation invariance, we applied a rotation matrix to the variables, $\boldsymbol{x} \mapsto B\boldsymbol{x}$, $B \in \mathrm{SO}(d, \mathbb{R})$. This is done for every benchmark function, and a rotation matrix was chosen randomly at the beginning of each trial. All starting points were drawn uniformly from $[0, 1]$, except for Sphere, where we sampled from $\mathcal{N}(0, \mathrm{I})$. For each function, we vary $d \in \{4, 8, 16, \dots, 256\}$. Due to the long running times, we only compute CMA-ES-Ref up to $d = 128$. For the given range of dimensions, for every choice of $d$, we ran 100 trials from different initial points and monitored the number of iterations and the wall-clock time needed to sample a point with a function value below $10^{-14}$. For Rosenbrock we excluded the trials in which the algorithm did not converge to the global optimum.

We further evaluated the algorithms on additional benchmark functions inspired by Stich and Müller [2012] and measured the change of rotation introduced by the Cholesky-CMA-ES at each iteration ($E_t$), see supplementary material.

**Results.** Figure 1 shows that CMA-ES-Ref and Cholesky-CMA-ES required the same amount of function evaluations to reach a given objective value. The CMA-ES/d required slightly more evaluations depending on the benchmark function. When considering the wall-clock runtime, the Cholesky-CMA-ES was significantly faster than the other algorithms. As expected from the theoretical analysis, the higher the dimensionality the more pronounced the differences, see Figure 2 (note logarithmic scales). For $d = 64$ the Cholesky-CMA-ES was already 20 times faster than the CMA-ES-Ref. The drastic differences in runtime become apparent when inspecting single trials. Note that for $d = 256$ the matrix size exceeded the L2 cache, which affected the performance of the Cholesky-CMA-ES and Suttorp-CMA-ES. Figure 3 plots the trials with runtimes closest to the corresponding median runtimes for $d = 128$.

## 5 Conclusion

CMA-ES is a ubiquitous algorithm for derivative-free optimization. The CMA-ES has proven to be a highly efficient direct policy search algorithm and to be a useful tool for model selection in supervised learning. We propose the Cholesky-CMA-ES, which can be regarded as an approximation of the original CMA-ES. We gave theoretical arguments for why our approximation, which only affects the global step-size adaptation, does not impair performance. The Cholesky-CMA-ES achieves a better, asymptotically optimal time complexity of $O(\mu d^2)$ for the covariance update and optimal memory complexity. It allows for numerically stable computation of the inverse of the Cholesky factor in quadratic time and provides the eigenvalues of the covariance matrix without additional costs. We empirically compared the Cholesky-CMA-ES to the state-of-the-art CMA-ES with delayed covariance matrix decomposition. Our experiments demonstrated a significant increase in optimizaton speed. As expected, the Cholesky-CMA-ES needed the same amount of objective function evaluations as the standard CMA-ES, but required much less wall-clock time – and this speed-up increases with the search space dimensionality. Still, our algorithm scales quadratically with the problem dimensionality. If the dimensionality gets so large that maintaining a full covariance matrix becomes computationally infeasible, one has to resort to low-dimensional approximations [e.g., Loshchilov, 2015], which, however, bear the risk of a significant drop in optimization performance. Thus, we advocate our new Cholesky-CMA-ES for scaling up CMA-ES to large optimization problems for which updating and storing the covariance matrix is still possible, for example, for training neural networks in direct policy search.

**Acknowledgement.** We acknowledge support from the Innovation Fund Denmark through the projects "Personalized breast cancer screening" (OK, CI) and "Cyber Fraud Detection Using Advanced Machine Learning Techniques" (DRA, CI).

## Footnotes

[1]Given $c_c$, the factors in (1) are chosen to compensate for the change in variance when adding distributions. If the ranking of the points would be purely random, $\sqrt{\mu_{\text{eff}}} \cdot (\boldsymbol{m}_{t+1} - \boldsymbol{m}_t)/\sigma_t \sim \mathcal{N}(0, C_t)$ and if $C_t = I$ and $\boldsymbol{p}_{c,t} \sim \mathcal{N}(0, \text{I})$ then $\boldsymbol{p}_{c,t+1} \sim \mathcal{N}(0, \text{I})$.

[2]Actually, the complexity is related to the complexity of multiplying two $\mu \times d$ matrices. We assume a naïve implementation of matrix multiplication. With a faster multiplication algorithm, the complexity can be reduced accordingly.

[3]We added our algorithm to the open-source machine learning library Shark [Igel *et al.*, 2008] and used LAPACK for high efficiency.

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
