[Supplementary Material · KrauseEtAlSupplement.pdf]

# CMA-ES with Optimal Covariance Update and Storage Complexity
## Supplementary Material

**Oswin Krause**
Dept. of Computer Science
University of Copenhagen
Copenhagen, Denmark
oswin.krause@di.ku.dk

**Dídac R. Arbonès**
Dept. of Computer Science
University of Copenhagen
Copenhagen, Denmark
didac@di.ku.dk

**Christian Igel**
Dept. of Computer Science
University of Copenhagen
Copenhagen, Denmark
igel@di.ku.dk

## A    Testing of Different Eigenvalue Spectra

(a) Logarithmic Spectra

(b) Iterations

Figure 1: Left: Spectra of the 9 test functions in log-domain. Blue distributions have more eigenvalues close to $\alpha = 0.5$ corresponding to more flat distributions, while Red distributions are more extreme distributions of eigenvalues. Right: Runtime of the algorithms on the functions ordered by difficulty of the distribution.

To further test the behaviour of the algorithms on quadratic functions with different eigenvalue spectra, we implemented an approach inspired by Stich and Müller [2012]. Starting from the spectrum of the ellipsoid function with eigenvalues spaced uniformly on log-scale, we implemented distributions which have either more eigenvalues that are small or large, forming more *extreme* distributions, or distributions with many eigenvalues that are close, forming *flat* distributions. The biggest difference to the approach by Stich and Müller [2012] is that we distribute the eigenvalues on a log-scale. The extreme distributions can be considered as difficult for optimization, because many more eigenvalues have to be learned in order to adapt to the conditioning of the problem, while in the flat cases only a few eigenvalues are relevant.

All functions have the form $f(\boldsymbol{x}) = \sum_{i=1}^{d} 10^{6\alpha_i} x_i^2$. where $0 = \alpha_0 \leq \alpha_2 \leq \cdots \leq \alpha_d = 1$ describe the eigenvalue distribution on log-scale. The extreme distributions follow

$$\alpha_i = \frac{\sigma(a(1 - 2 \cdot i/d)) - \sigma(a)}{\sigma(-a) - \sigma(a)} \quad ,$$

with $\sigma(x) = 1/(1 + \exp(-x))$ and the flat distributions follow

$$\alpha_i = \frac{\beta_a(i) - \beta_a(1)}{\beta_a(d) - \beta_a(1)} \quad,$$

where $a > 0$ is a variable governing the shape of the distribution and

$$\beta_a(i) = \log\left(\frac{1}{10^{-a} + i/d \cdot (1 - 2 \cdot 10^{-a})} - 1\right) \quad.$$

As $a \to 0$, both distributions approach the uniform distribution leading to the Ellipsoid. The extreme distributions have sigmoidal shapes, while flat distributions have shapes similar to the inverse of a sigmoid. The values of $a$ used in the extreme distributions were $2, 5, 8, 15$ and for the flat $1, 1.25, 2, 3, 6$. The resulting distributions of $\alpha_i$ can be seen in Figure 1a. We ran 51 trials with 64 dimensions until a target value of $10^{-14}$ was reached and show the median of the number of iterations in Figure 1b.

We can see that all approaches except CMA-ES/d showed comparable performance on all spectra, where for the harder spectra the gap increased, clearly showing a loss in performance when skipping updates. Suttorp-CMA-ES and Cholesky-CMA-ES showed comparable results.

## B    Error of the Rotation Matrix

Figure 2: Frobenius distance $\|E_{t+1}E_t^T - I\|_F$. The line marking the distance of a single rotation by $0.125\,\mathrm{rad}$ is given as a reference.

This section describes experiments studying the change of the rotation matrix $E_t$ introduced by the Cholesky-CMA-ES at each iteration. As described in the main paper, a constant rotation matrix $E_t$ will not change the behaviour of the algorithm, but only the rotation introduced between iterations, $E_{t+1}E_t^T$. To measure this, we use the Frobenius distance from the identity matrix $\|E_{t+1}E_t^T - I\|_F$ during a run on DiffPowers. As a reference, we calculated an upper bound on a single rotation angle that results in a rotation matrix with similar Frobenius distance. In reality the rotation angles of the matrix are typically smaller as rotations are split among dimensions.

## References

S. U. Stich and C. L. Müller. On spectral invariance of randomized Hessian and covariance matrix adaptation schemes. In *Parallel Problem Solving from Nature (PPSN XII)*, pages 448–457. Springer, 2012.