[Reviews · NeurIPS 2016]

Reviewer 1

Summary

CMA-ES is the state-of-the art stochastic/randomized algorithm for (derivative-free) optimization. It combines different learning mechanisms for adapting the parameters of a multivariate normal sampling distribution that can be equivalently seen as adaptation of an underlying metric. This paper proposes a new variant for the update of the covariance matrix based on triangular Cholesky factors that has the advantage compared to previous updates to significantly reduce the complexity of the update as well as the memory needed to store the state parameters of the algorithm.

Qualitative Assessment

The new method is carefully evaluated numerically. It is shown that the new update does not deteriorate the performance of the standard CMA-ES in terms of number of function evaluations to reach a given target while it significantly reduce the wall-clock time of the algorithm, particularly for large dimensions. I think that this paper is a very significant contribution in the domain of optimization. I expect that this variant of CMA-ES can become a default variant of the method in the context of medium/large-scale as it has several clear advantages over the current default method: namely reduced memory needed, reduced complexity and getting for free the eigenvalues of the covariance matrix. The paper is nicely written and the approach carefully evaluated. It is nice to see this contribution at NIPS given that the CMA-ES is strongly connected to machine learning, information geometry and is also used in reinforcement learning or supervised learning. Minor: explain how you ensure that the different implementations used are comparable in terms of CPU (is it the same implementation language, which one, …) When commenting on Figure 2, it is written “the higher the dimensionality, the more pronounced the differences”, it does not seem to be the case for Cigar and Discus in dimension 256. It would be good to provide an explanation for that.

Confidence in this Review

3-Expert (read the paper in detail, know the area, quite certain of my opinion)


Reviewer 2

Summary

This paper adopts the Cholesky decomposition instead of the eigen-decomposition to reduce the time and storage complexity of the CMA-ES.

Qualitative Assessment

Generally speaking, this paper is clearly written. The proposed method seems reasonable and the experimental results (especially the speedup rates) are quite promising. I think the proposed method may become a new standard version of CMA-ES in the future. Although this paper can be meaningful, the originality is limited because I think this work is a trivial extension of the previous work: A More Efficient Rank-one Covariance Matrix Update for Evolution Strategies. The methods are basically identical. Please correct me if I am wrong. Besides, some very important contents are strangely missing. 1) You should show whether your algorithm is equivalent to formula (2). I understand the space is limited and it has been proven in the previous work, but this is still essential for this paper. 2) You claim that the computational complexity is O(\mu d^2), which is exactly the same as CMA-ES-Ref. Please discuss in detail (maybe using the '\Theta' notation instead of the 'O') why your algorithm is faster. 3) You claim that the error in the path is small and decrease over time, which is contrary to Hansen's claim. Since your theoretical analysis is not deterministic, experiments should be conducted to validate your claim. Comparing only the final performance is not sufficient. Some smaller comments: 1) Please check your notations carefully. For example, w_k in Algorithm 2 should be \alpha_k, n in Table 1 should be d. 2) 'F' should be 'f' in line 185. As such, I think this paper is not qualified for NIPS in its current form.

Confidence in this Review

2-Confident (read it all; understood it all reasonably well)


Reviewer 3

Summary

This paper considers Cholesky covariance matrix adaptation evolution strategy with both rank-1 and rank-\mu update for triangular Cholesky factor (in sequence of rank-1 update). This paper proofed, under the convergence assumption of the covariance, that the modified cumulative step-size adaptation does not make significant change of the algorithm behavior. The paper presents experiments on algorithm performances.

Qualitative Assessment

1. This paper considers the Cholesky CMA-ES. It extends the update for triangular Cholesky factor using both rank-1 and rank-\mu update. Yet, I don't think its contribution is significant, considering the work of Krause et al. 2015 which uses rank-1 update to triangular Cholesky factor. This paper presents a straightforward generalization of this procedure to rank-\mu update in a sequences of rank-1 updates. 2. Generally, the convergence of covariance $C_t$ in CMA-ES is proofed by theoretical analysis (Beyer, ECJ, 2014), although it is not based on the exact formulae of CMA-ES. This may improve the confidence of Lemma1. 3. There are some other problems. 3.1. In equation (1), the \sigma in denominator should be \sigma_t. The same problem occurs in the footnote 2 on page 3. 3.2. In line 94, 'which can by be motivated by principles from ...' , the world 'by' should be deleted.

Confidence in this Review

2-Confident (read it all; understood it all reasonably well)


Reviewer 4

Summary

This work’s main contribution is to utilize the triangular Cholesky factorization technique which could get rid of the square root in the standard CMA-ES, which could save the computation cost when calculating the inverse of matrices. The author presents an approach that comes with a theoretical justification to illustrate that it does not deteriorate the algorithm’s performance. The proposed method reduces the runtime complexity of the algorithm with no significant change in the number of objective function evaluations.

Qualitative Assessment

This paper is a valuable implementation of Cholesky factorization method for CMA-ES. Can the author further explain theoretically why the ‘numerically more stable’ computation could be achieved? It seems that The Cholesky-CMA-ES needs the same amount of objective function evaluations as the standard CMA-ES, and it lessen the wall-clock time and this increases with the search space dimensionality. Could the author consider combining the low-dimensional approximation [Loshchilov, 2015] with the existing framework because in most of practical cases such as training neural networks in direct policy search, the high-dimensionality of the problems should be a non-neglected concern?

Confidence in this Review

2-Confident (read it all; understood it all reasonably well)


Reviewer 5

Summary

The manuscript describes a novel CMA-ES update scheme, the Cholesky-CMA-ES, that approximates the original covariance matrix adaptation (with complexity O(d^3)) by a novel scheme that runs in O(\mu d^2) where d is the optimization problem dimension and \mu the number of samples per iterations (typically O(log(d)). The principle idea of the update is to maintain and update a lower-triangular Cholesky matrix rather than a full covariance matrix (and corresponding sqrt of that). The difference between the Cholesky and the original CMA-ES is due to the lack of estimating a (random) orthogonal matrix E_t in every step. Under quite restrictive assumptions (i.e., assumptions about convergence of a sequence of covariance matrices to an unknown target covariance matrix) the authors give an asymptotic argument that the sequence of E_t will converge to an identity matrix, and thus the two schemes become equivalent. The validity and the general speed-up of the Cholesky-CMA-ES is illustrated by using standard runtime and convergence plots on selected objective functions and by comparing it to two state-of-the-art CMA-ES alternatives that delay the full covariance matrix update by different means.

Qualitative Assessment

This paper makes a solid contribution to speeding up the celebrated CMA-ES algorithm which is arguably the best algorithm for non-convex black-box optimization when little is known about the topology of the objective function and first or higher-order information is difficult to access or not helpful. The proposed method speeds up the algorithm considerably and shows no deterioration of optimization performance on the synthetic test cases. Although this is of general importance, one weak point of ANY speed-up proposal for CMA-ES is that, in practice, the overall optimization process is dominated by the runtime of the black-box oracle (the objective function), and, often, the actual runtime of the CMA-ES is negligible compared to that. In addition, in typical scenarios where CMA-ES is used, the dimensionality of the problems does not go beyond a few tens of variables. Only in the case where a single function evaluation is on the order of milliseconds or smaller and the dimensionality is higher, the runtime of CMA-ES plays an important role, and this is targeted scenario here (which may be less realistic in practice). As theoretical guarantees are notoriously difficult to get for an involved scheme such as CMA-ES, I would like to propose a couple of further empirical experiments and plots that may strengthen the presentation of the results. First, it may be instructive to see the evolution of E_t on selected optimization trajectories, e.g., in different norms. I envision an experiment where the full CMA-ES is used and the approximation is calculated simultaneously and compared. This would give further empirical evidence about the actual fluctuations of E_t over time. Second, it may be instructive to see actual runtimes and performance of CMA-ES without delay strategies in this contribution for completeness. Third, although the used test functions are standard in the evolutionary optimization community I would prefer, on quadratic functions, a simulation study where the spectrum of quadratic forms is varied continuously (see, e.g., Stich and Mueller, PPSN 2012), and compare the solution quality and error in E_t in dependence of the spectral distribution. Forth, it would be instructive to see a real-world experiment that accompanies the present simulation results. For instance, policy gradients in real-time robotics applications might be a good choice where speed indeed matters and the parameter space is not too large.

Confidence in this Review

3-Expert (read the paper in detail, know the area, quite certain of my opinion)


Reviewer 6

Summary

The paper proposes a version of CMA-ES with "Optimal Covariance Update and Storage Complexity". The authors suggest that the proposed modification does not change the behavior of the algorithm but makes it cheaper to run.

Qualitative Assessment

The proposed method is called Cholesky-CMA-ES and this is misleading because this name was first introduced by T. Suttorp et al. 2009 for a similar algorithm and is well known. The authors do not mention this issue, instead they introduce their variant as "Our CMA-ES variant, referred to as Cholesky-CMA-ES". The proposed method is an incremental modification to the work of Krause et al. 2015. More specifically, the only algorithmic contribution is to call rankOneUpdate (it is also present in the work of Krause et al.) not once as in Plus strategies but in a loop for all individuals. Empirically, there is a difference on Cigar function in Figure b): the observed plateau has different objective function values for CMA-ES-Ref and Cholesky-CMA-ES. It might because of the way the authors plot median results. The paper should compare the proposed approach to Cholesky-CMA-ES by Suttorp (empirically) and Cholesky-CMA-ES by Krause (algorithmically). The claim about "Optimal Covariance Update and Storage Complexity" is a bit misleading. It is not optimal when we consider Sphere function where we don't need the covariance matrix at all, on separable functions we may deal with the diagonal only. The proposed approach is only two times cheaper than the omitted Cholesky CMA-ES by Suttorp. The latter is claimed in this paper to be prone to numerical instabilities but the original paper suggested the opposite. Again, its omission in Figure 3 can mislead the reader regarding the actual contribution of the paper. I don't think that the proposed approach changes something for CMA-ES regarding large-scale optimization since the complexity is still quadratic. However, I admit that the proposed modification might be the best thing one can do to save time and memory when the full covariance matrix is considered. Overall, I think that the work is too incremental (algorithmically, it is only one "for" loop). I suggest the following modifications: a) Resolve the naming issue; b) Perform experiments with variants of Suttorp et al. and Krause et al. and update all 3 figures. If they will become unreadable, then shown only the best performing variant in the main text and show the full figures in Supplementary Material; c) Better explain that the main algorithmic contribution is the "for" loop in Algorithm 1 or explain why it is wrong to think so. Minor notes: "Typically, an eigenvalue decomposition of the covariance matrix is performed in each update step." This is atypical, i.e., in all proper implementations it is done periodically.

Confidence in this Review

2-Confident (read it all; understood it all reasonably well)